## RESEARCH CULTURE

# A survey of new PIs in the UK

**Abstract** The challenges facing a new independent group leader, principal investigator (PI) or university lecturer are formidable: secure funding, recruit staff and students, establish a research programme, give lectures, and carry out various administrative duties. Here we report the results of a survey of individuals appointed as new group leaders, PIs or university lecturers in the UK between 2012 and 2018. The concerns expressed include difficulties in recruiting PhD students, maintaining a good work-life balance and securing permanent positions. Gender differences were also found in relation to starting salary and success with research funding. We make recommendations to employers and funders to address some of these concerns, and offer advice to those applying for PI positions.
DOI: https://doi.org/10.7554/eLife.46827.001

**SOPHIE E ACTON\*, ANDREW JD BELL\*, CHRISTOPHER P TOSELAND\* AND ALISON TWELVETREES\***

**\*For correspondence:** s.acton@ucl.ac.uk (SEA); andrew.j.d.bell@sheffield.ac.uk (AJDB); c.toseland@sheffield.ac.uk (CPT); a.twelvetrees@sheffield.ac.uk (AT)

**Competing interests:** The authors declare that no competing interests exist.

## Introduction

Academic careers have expanded across the university sector at all career levels in recent decades, but there remain relatively few levels in the hierarchy – PhD student, postdoctoral researcher, and independent group leader/principal investigator (PI)/university lecturer. In general the system trains more PhDs and postdocs than can be employed as independent group leaders, PIs or university lecturers, so the competition for these positions can be intense. The present authors know this first-hand because we all started in such positions at universities in the UK between 2015 and 2018. Moreover, we know that new group leaders, PIs and university lecturers face a wide range of challenges as they seek to establish their research groups and undertake new responsibilities in their department or institute.

In this article we will use the term PI as a short-hand for independent group leader, principal investigator or university lecturer. There are several routes to becoming a PI in the UK. The two most frequent are: i) recruitment as a permanent lecturer (subject to passing probation) at a university; ii) recruitment as a fixed-term research fellow at a university or research institute (funded directly by a university, or externally by research councils and charities). There are pros and cons associated with both routes: lecturers typically have long-term job security, but externally-funded research fellows can often establish their research programme faster than lecturers.

To explore how new PIs are recruited, supported and evaluated, we conducted a survey of PIs appointedin the UK between 2012 and 2018. This article reports the results of this survey, discusses what it tells about the hopes and concerns of new PIs in the UK, makes recommendations to employers and funders to address some of these concerns, and concludes with advice for those hoping to become a new PI.

## Results

Our survey is described in detail in the Methods section. In summary, the survey was completed by 365 individuals who had become PIs over the past six years: 83% worked in the life sciences (*Figure 1A*), 57% were male (*Figure 1B*), 51% were from the UK (*Figure 1C*), and 84% classified themselves as white (*Figure 1D*).

The average respondent had spent seven years between finishing their PhD and starting their own group (*Figure 1E*), which suggests they had held two or three fixed-term positions before gaining independence. This seven-year period likely reflects the eligibility restrictions

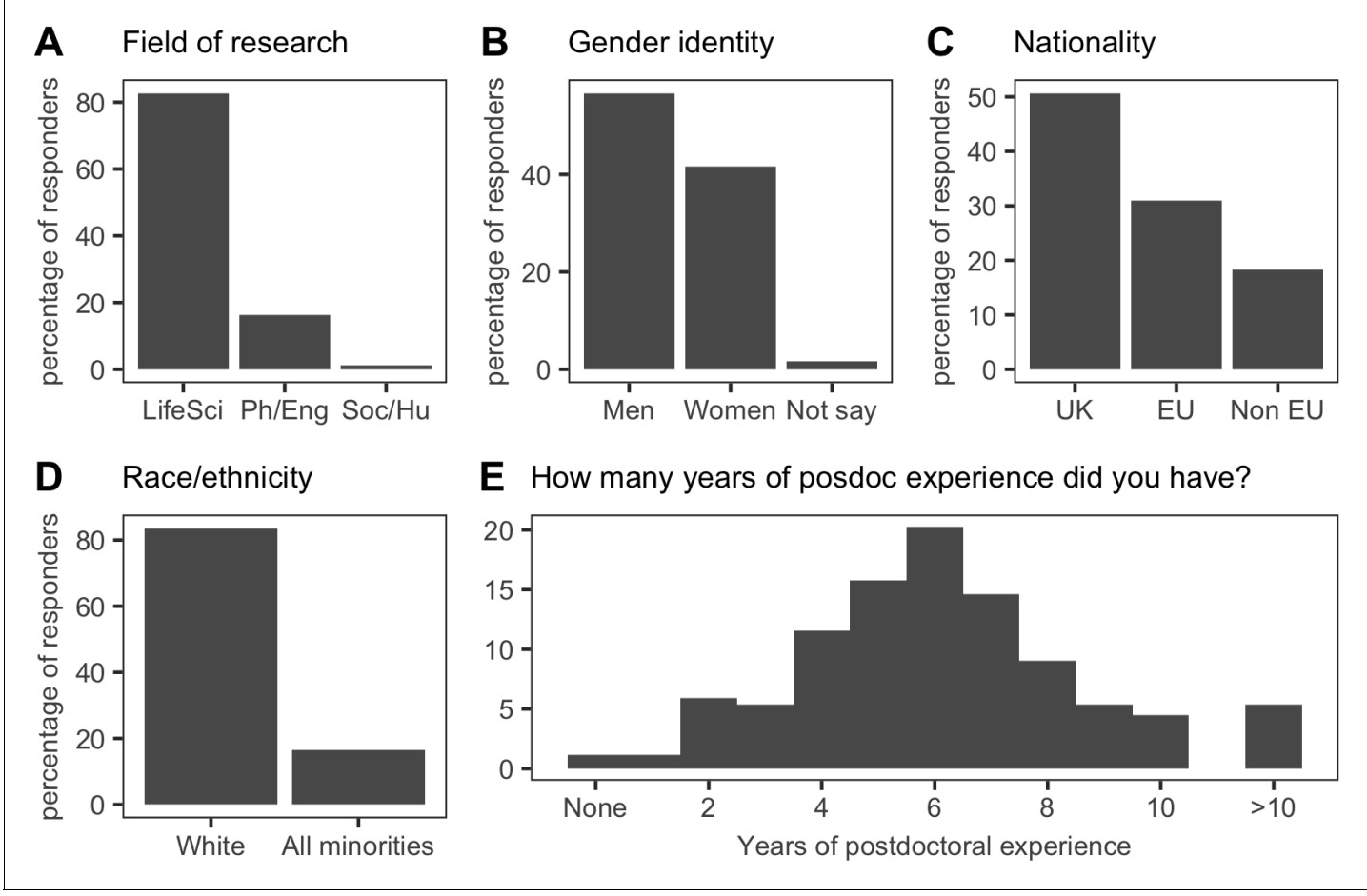

**Figure 1.** Overview of cohort demographics. (**A**) 83% (302/365) of respondents classified themselves as being in the Life Sciences (LifeSci); 16% (59/365) were in the Physical Sciences and Engineering (Ph/Eng); and 1% (4/365) were from in the Social Sciences and Humanities (Soc/Hu). (**B**) 56.7% (207/365) of respondents were men; 41.6% (152/365) were women; and 1.6% (6/365) were 'Prefer not to say'. (**C**) 51% (185/365) of respondents were from the UK; 31% (113/365) were from other EU countries; and 18% (67/365) were from the rest of the world (Non EU). (**D**) 84% (305/365) of respondents were white. (**E**) 51% (180/355) of respondents had between five and seven years of postdoc experience prior to independence. Consequently, the majority of respondents were in their mid-thirties at the time they became new PIs (see *Figure 1—figure supplement 1*); the most recent new PIs were the least likely to have dependents (see *Figure 1—figure supplement 2*).

DOI: https://doi.org/10.7554/eLife.46827.002

The following source data and figure supplements are available for figure 1:

**Source data 1.** Summary data for *Figure 1*.
DOI: https://doi.org/10.7554/eLife.46827.007
**Figure supplement 1.** Age of respondents.
DOI: https://doi.org/10.7554/eLife.46827.003
**Figure supplement 1—source data 1.** Summary data for *Figure 1—figure supplement 1*.
DOI: https://doi.org/10.7554/eLife.46827.004
**Figure supplement 2.** Dependents, career breaks and work patterns.
DOI: https://doi.org/10.7554/eLife.46827.005
**Figure supplement 2—source data 1.** Summary data for *Figure 1—figure supplement 2*.
DOI: https://doi.org/10.7554/eLife.46827.006

that were in place for some fellowships until recently, and it will be interesting to see if the recent decision by many funders to remove such restrictions has an influence on the time taken for PhDs to become PIs.

A longer time spent as a postdoctoral researcher may allow some individuals the extra time needed to complete and publish ambitious or collaborative projects. However, it might also increase the average age at which researchers

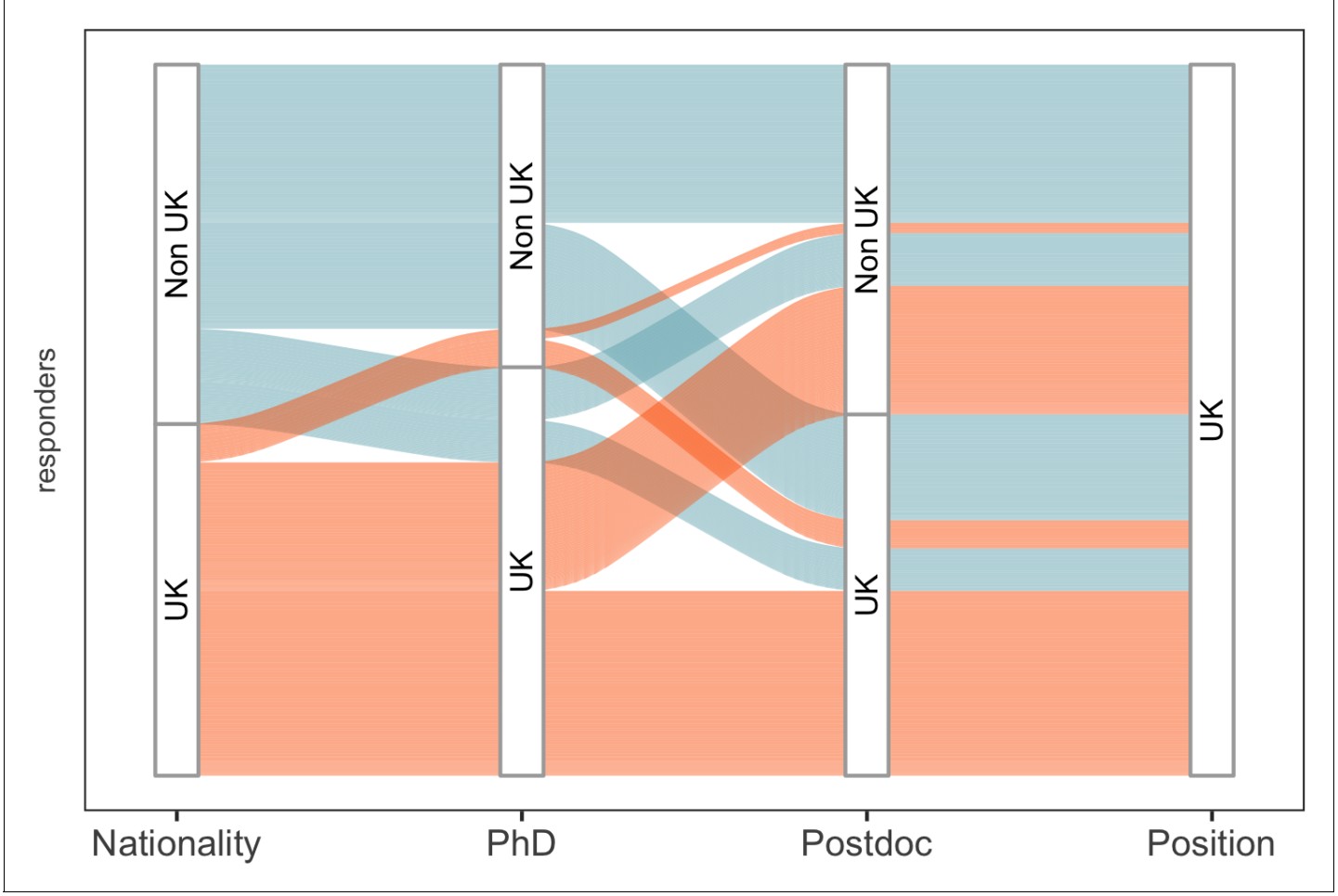

**Figure 2.** Overview of cohort mobility. An alluvial plot of cohort migration, where line width is proportional to the percentage of respondents. Colour corresponds to the nationality (UK; orange, Non-UK; blue) of participants as they move through their careers (PhD and postdoc training). 26.8% (98/365) of respondents had spent all their career in the UK, while 22.2% (81/365) had not worked in the UK before starting as a new PI in the UK.

DOI: https://doi.org/10.7554/eLife.46827.008

The following source data is available for figure 2:

**Source data 1.** Summary data for *Figure 2*.

DOI: https://doi.org/10.7554/eLife.46827.009

become independent (which is 34 years old for the respondents to our survey; *Figure 1—figure supplement 1*), and this would increase the length of time during which postdocs have to balance starting a family with the pressures imposed by fixed-term contracts and the need to show mobility.

With regard to families, half of our cohort had dependents, female investigators had taken the longest career breaks and were the most likely to have dependants, and almost everyone (361/365) worked full time (*Figure 1—figure supplement 2*).

International mobility plays a key role in the academic career path. 51% of respondents had spent more than one year training outside of the UK as postdocs, and 67% had undertaken at least one international move as part of their career (either when moving between their PhD and postdoc, or when moving from a postdoc to a PI position in the UK; *Figure 2*). And when moves within the UK are included, the percentage of respondents who moved to take up their position increases to 78%: moreover, 77% had not previously worked in their current department.

### Job satisfaction and well-being

While details about our cohort reveal information about the sector as a whole, it is when we ask about job satisfaction that we begin to see where problems may lie for new PIs. In general

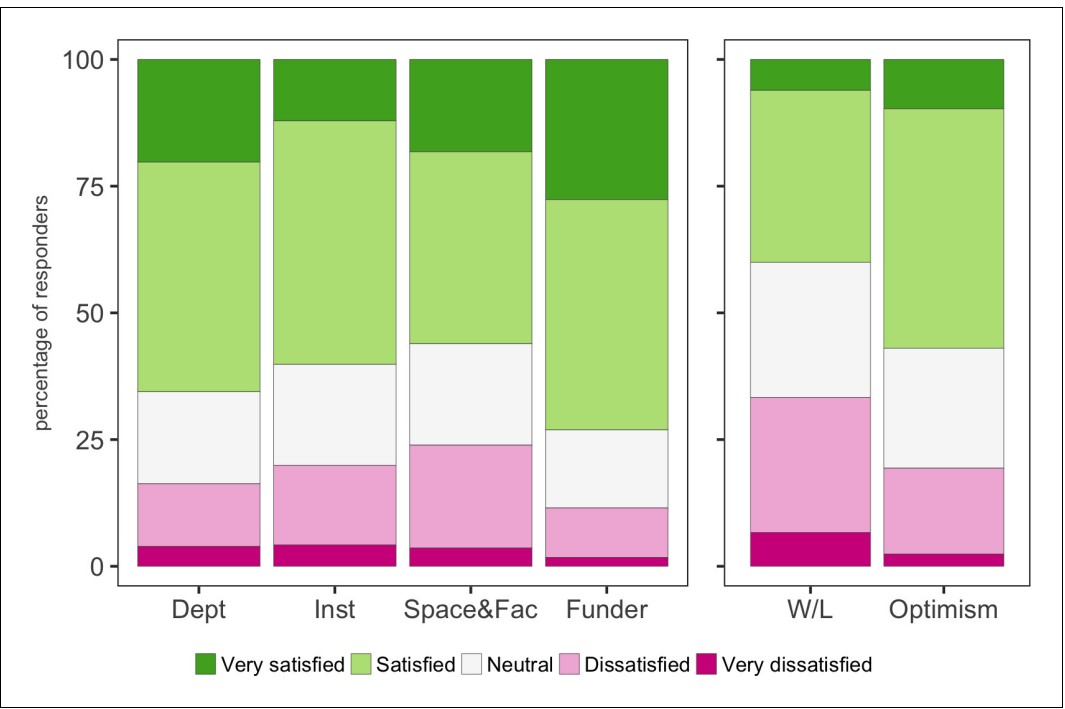

**Figure 3.** Satisfaction and optimism. Participants were asked to rate their satisfaction with their host department (Dept), host institution (Inst), lab space and access to facilities (Space&Fac) and support from their funder (Funder). Participants were also asked how they felt about their current work-life balance (W/L) and their optimism about their future career (Optimism). With the exception of work-life balance, more than 50% of respondents replied that they were satisfied or very satisfied.

DOI: https://doi.org/10.7554/eLife.46827.010

The following source data and figure supplements are available for figure 3:

**Source data 1.** Summary data for *Figure 3*.
DOI: https://doi.org/10.7554/eLife.46827.013
**Figure supplement 1.** Optimism and work-life balance of subgroups.
DOI: https://doi.org/10.7554/eLife.46827.011
**Figure supplement 1—source data 1.** Summary data for *Figure 3—figure supplement 1*.
DOI: https://doi.org/10.7554/eLife.46827.012

the number of respondents who replied that they were satisfied or very satisfied clearly exceeded the number who were dissatisfied or very dissatisfied about the six different aspects they were asked about: their department; their institution; space and facilities; their funder; work-life balance; and optimism (*Figure 3*). However, 24% (79/330) were either dissatisfied or very dissatisfied with their space and facilities, compared with 56% (185/330) who were satisfied or very satisfied. We think there is scope for funders to put pressure on institutions to address concerns about space and facilities.

Moreover, 33% (111/331) were either dissatisfied or very dissatisfied with their work-life balance, compared with 41% (135/331) who were satisfied or very satisfied. Flexible or part-time working is often put forward as a way to improve

work-life balance but, as noted above, just four of the respondents worked part-time. Given all the challenges associated with being a new PI – find funding, build a research group, publish work, prepare and give lectures – it is not surprising that practically no one works part-time.

The pressures of being a new PI are best expressed in this direct quotation from one respondent: "I feel like I'm trying to do three separate jobs (research, management/admin, teaching) as well as be a mother... be my own postdoc (because I can't afford one), be the lab technician (because I can't afford one), be the lab manager (because I can't afford one...), be a good mentor for my students, plan strategy, write grants (constantly, I need the money), stay up to date with other research, prepare new teaching material (this takes me ages, I want to

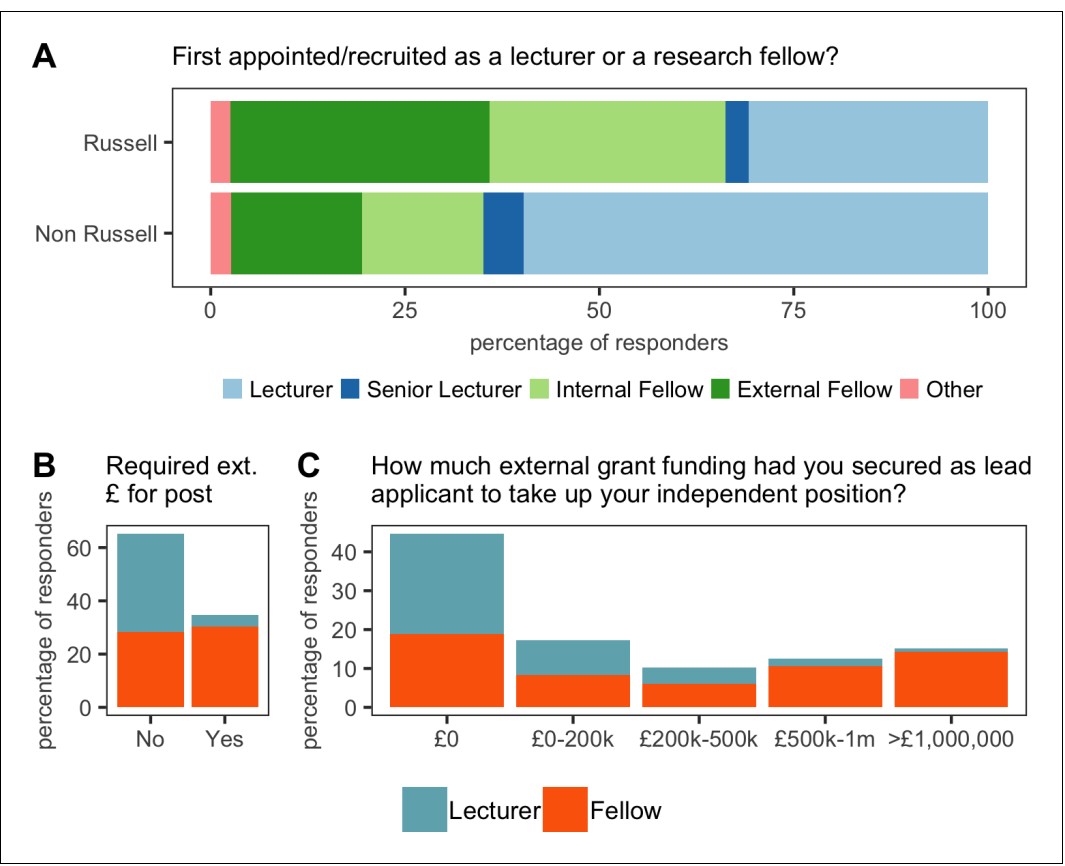

**Figure 4.** Initial recruitment. (**A**) 75% (234/311) of respondents were from Russell Group universities, and a majority of these recruits (64%; 149/234) were brought in as research fellows (top). 25% (77/311) of respondents were from outside the Russell Group, and a majority of these recruits (60%; 46/77) were brought in as lecturers (bottom). (**B**) 35% (108/311) of respondents were required to have secured a major grant or fellowship in order to take up their position: 52% (94/182) of fellows were expected to have secured such funding, compared with just 11% (14/129) of lecturers. (**C**) Some respondents (mostly research fellows) had secured more than £1 m in external grant funding when they started as a new PI.

DOI: https://doi.org/10.7554/eLife.46827.014

The following source data is available for figure 4:

**Source data 1.** Summary data for *Figure 4*.
DOI: https://doi.org/10.7554/eLife.46827.015

do a good job), teach, mark assessments and answer student queries etc. I could go on. No, seriously, is it even possible?".

Clearly there is a need to find other mechanisms to improve the work-life balance besides offering part-time/flexible working patterns. However, despite elements of dissatisfaction, it is important to highlight over 50% of respondents were optimistic for the future (*Figure 3*). This highlights a strong resilience and positivity amongst new PIs as they tackle the various demands of their role. It was also encouraging to find that having dependants did not affect satisfaction ratings or optimism scores for new male or female PIs (*Figure 3—figure supplement 1*).

### Career track comparison and gender disparity

As previously mentioned, there is no single route to becoming a PI in the UK. Typically, two career paths exist: (a) appointment as a lecturer, possibly followed by promotion to senior lecturer, then reader and lastly professor; (b) appointment as a research fellow, possibly followed by an advanced fellowship or a transfer to the lecturer career path. 41% of our respondents were on the lecturer career path, with 59% on the research fellow path (*Figure 4*). The research

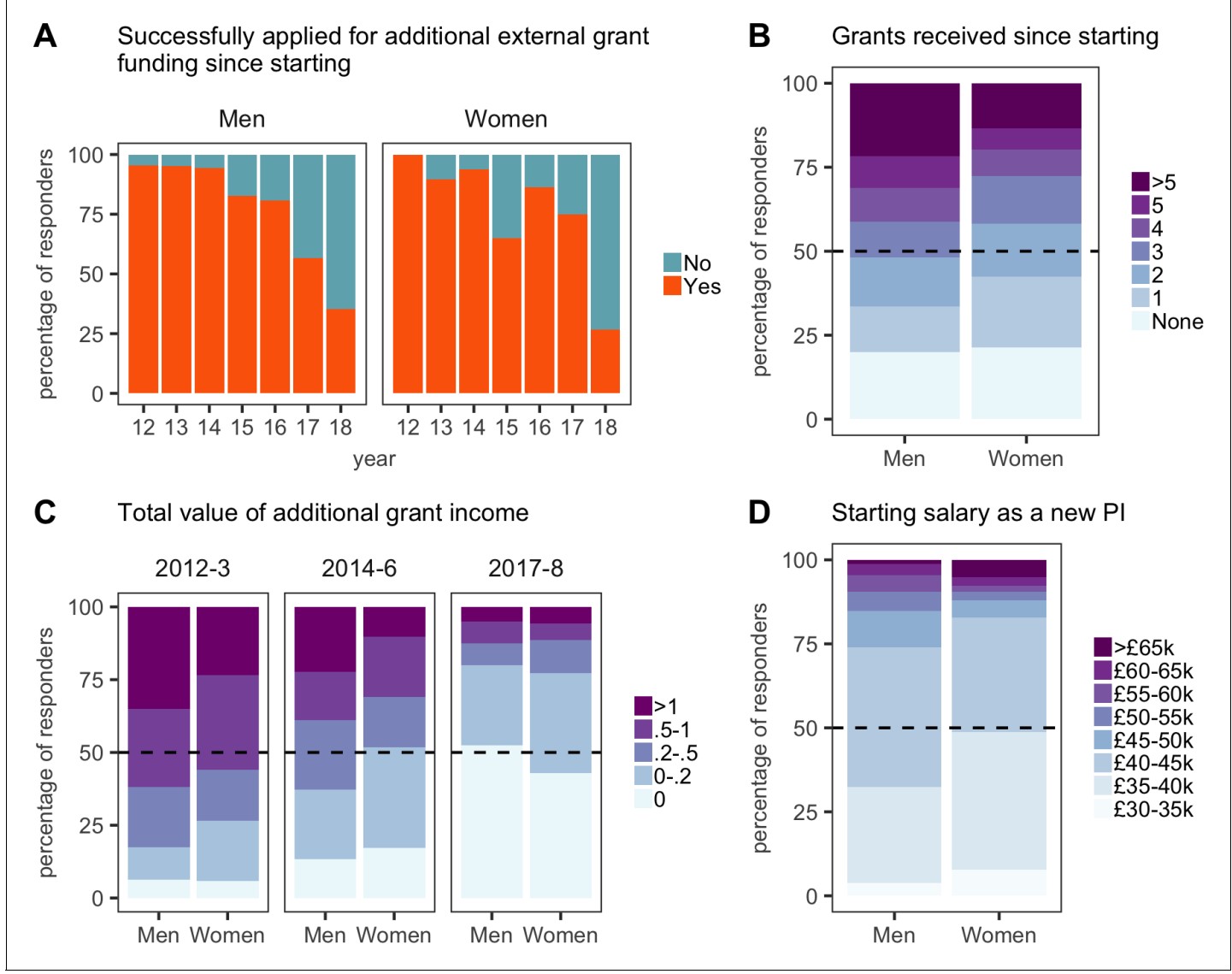

**Figure 5.** Gender comparisons in pay and grant income. All plots are expressed as the percentage of respondents within each category. (**A**) Grant success versus year of independence (12 = 2012, and so on). (**B**) Half of the male respondents had received three or more grants since starting; half of the female respondents had received two or more. (**C**) Grant values (expressed in £m) for new PIs who started in 2012–13, 2014–16, and 2017–2018. (**D**) The self-reported salaries of new PIs at the time they were appointed show a substantial gender pay gap.

DOI: https://doi.org/10.7554/eLife.46827.016

The following source data and figure supplements are available for figure 5:

**Source data 1.** Summary data for *Figure 5*.
DOI: https://doi.org/10.7554/eLife.46827.019
**Figure supplement 1.** Starting salaries.
DOI: https://doi.org/10.7554/eLife.46827.017
**Figure supplement 1—source data 1.** Summary data for *Figure 5—figure supplement 1*.
DOI: https://doi.org/10.7554/eLife.46827.018

fellows secured funding from a range of bodies and 70% of respondents were from 24 research-focused institutions, referred to as the Russell group (*Figure 4A*). This puts a large amount of resources into very few institutions.

38% of our new PIs were required to successfully apply for major grants or fellowships in order to take up their position (*Figure 4B*), with research fellows bringing in the highest levels of funding (some 25% of research fellows secured

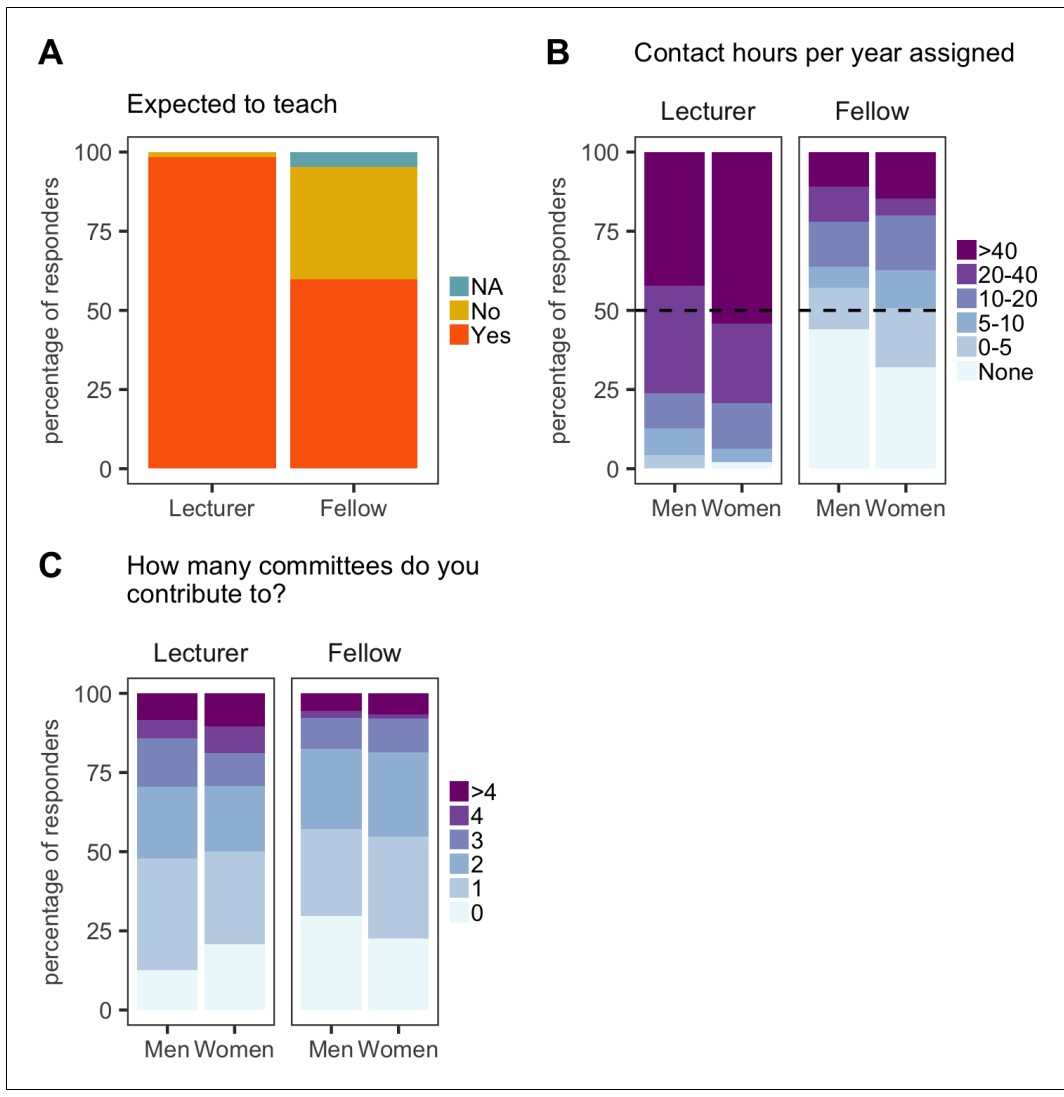

**Figure 6.** Teaching and administration load. All plots are expressed as the percentage of respondents within each category. (**A**) Almost all (119/121) new PIs appointed as lecturers are expected to teach, along with 60% (100/167) of new PIs appointed as fellows. (**B**) Lecturers generally have much higher teaching loads than fellows, and women have more contact hours assigned than men (both as lecturers and fellows). (**C**) Women were expected to contribute to more committees than men.

DOI: https://doi.org/10.7554/eLife.46827.020

The following source data is available for figure 6:

**Source data 1.** Summary data for *Figure 6*.

DOI: https://doi.org/10.7554/eLife.46827.021

more than £500,000 before starting their position; *Figure 4C*).

We also start to see a gender disparity emerge in grant funding early in the careers of new PIs: the majority of respondents (80% of men, 77% of women) had secured some additional funding within the first five years, but male respondents had secured significantly more than female respondents (*Figure 5A–C*). In particular, male PIs are much more likely to have secured

additional funding in excess of £1 m (p=0.025), and female PIs were awarded significantly fewer grants (p=0.039). Compared to new female PIs, it looks as if new male PIs were better able to gain momentum and accelerate through continued grant success, allowing them to build critical mass expanding the numbers in their labs. We should delve deeper into this issue and ensure that new female PIs are being encouraged and supported to apply for more funding and to

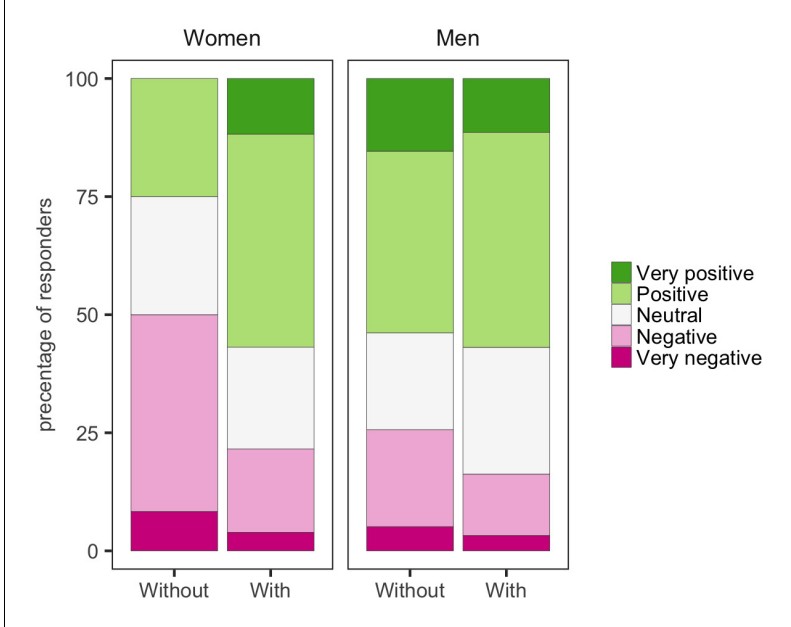

**Figure 7.** Mentorship and optimism. Women with mentors were more optimistic about the future than women without mentors. Men with mentors were a little more optimistic about the future than men without mentors.

DOI: https://doi.org/10.7554/eLife.46827.022

The following source data is available for figure 7:

**Source data 1.** Summary data for *Figure 7*.

DOI: https://doi.org/10.7554/eLife.46827.023

build their teams in the same way as new male PIs.

Female PIs were also paid £3000–5000 less per year than male PIs (*Figure 5D*), both as lecturers and research fellows (with lecturers generally starting on lower salaries than research fellows; *Figure 5—figure supplement 1A*): moreover, male PIs tended to be appointed at a higher grade (eg, at grade 8 rather than grade 7 for lectureships; *Figure 5—figure supplement 1B*). The appointment of women to lower grades will impact their rate of career progression compared to men.

### Teaching and administration

It came as a surprise to us that a majority of research fellows (57%) were expected to teach, despite their time being protected for research and a majority having their salary paid by an external funder rather than the university (*Figure 6A*). Although the number of contact hours was significantly less than the number for lecturers, nearly 40% of research fellows were expected to have more than 10 contact hours with undergraduates per year, and 10–15%, were expected to have more than 40 contact

hours per year (*Figure 6B*). There is an argument that research fellows should engage with their departments, bring new material to undergraduate courses, and participate in some level of teaching early in their independent careers: moreover, if the new PI hopes to be appointed as a lecturer in the longer term, gaining teaching experience will be beneficial. However, having to give more than 40 hr of lectures and tutorials is excessive for a new research fellow, so we suggest that all funders should consider specifying a limit on teaching hours. Some funders already apply limits, but these data suggest there is a lack of enforcement by either the fellow or the institution.

In general, we would suggest that the best way for new PIs to engage with undergraduate teaching would be to focus on the supervision of undergraduate laboratory projects or literature projects, so that the teaching is directly contributing to their research programme. After 2–3 years of having their time 'protected' in order to establish their research programme, direct teaching commitments through lectures and examinations can begin.

The fact that new female PIs have higher teaching loads (*Figure 6B*) and higher administration loads (*Figure 6C*) than new male PIs may contribute to the reduced grant success (and to the smaller lab sizes discussed below). These statistics suggest female PIs are over-committing themselves and/or are more frequently tasked with non-research roles than male PIs. Efforts by universities to ensure equal numbers of men and women sit on committees could contribute to the higher administration loads on female PIs (both new and established) because there are fewer female PIs overall.

### Mentorship and career development

Support for new PIs is especially important when they take on roles they have no prior experience of. We found that almost 25% of all new PIs felt that they had no mentorship: moreover, the female PIs who did not have a mentor were the least optimistic for their career progression (*Figure 7*). All but 3% of lecturers reported having an annual review, but almost 18% of externally funded fellows did not have an annual review. However, these fellows benefited from forms of support not available to most lecturers (for example, funders organize meetings that bring together all the fellows they fund, and these meetings can be a valuable source of peer support and career advice).

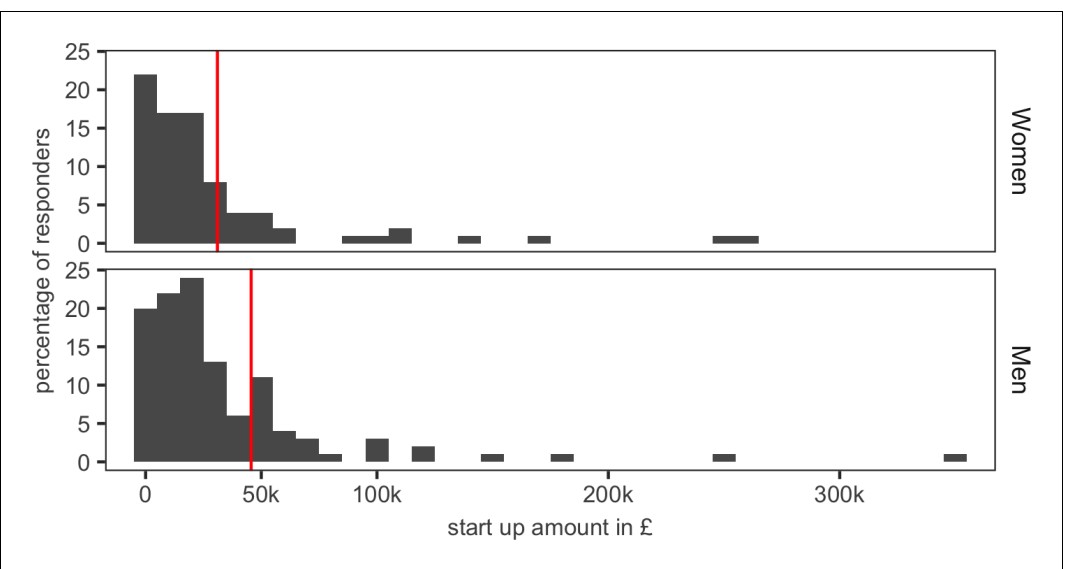

**Figure 8.** Start-up funds. Female respondents received an average of £31 k (red line; top panel) in start-up funds, whereas men received an average of £45.6 k (red line; bottom panel.

DOI: https://doi.org/10.7554/eLife.46827.024

The following source data and figure supplements are available for figure 8:

**Source data 1.** Summary data for *Figure 8*.
DOI: https://doi.org/10.7554/eLife.46827.027
**Figure supplement 1.** Proleptic appointments.
DOI: https://doi.org/10.7554/eLife.46827.025
**Figure supplement 1—source data 1.** Summary data for *Figure 8—figure supplement 1*.
DOI: https://doi.org/10.7554/eLife.46827.026

Most lecturers (88%) were provided with start-up funds, which were typically in the region of £20–60K, by their university (*Figure 8*). Once again there was a gender gap in favour of male PIs, who received an average of £14.6K more than female PIs.

A major concern for externally funded research fellows was what to do at the end of their contract. While one might assume that such a fellow would be appointed as a lecturer (or higher on the academic career ladder) at their host university when their fellowship ended, 70% of research fellows did not have an agreement for such a 'proleptic appointment' in place with their host university (*Figure 8—figure supplement 1*). Moreover, 58% were unaware of what they had to do to get a permanent position or how their host institution or department went about making such decisions. Since these fellows are hired by universities on contracts which are dependent on the funding source, they can be made redundant at the end of the fellowship with little consequence. These fellows do not have job security despite supporting their own salary, bringing in large grants, passing stringent external selection processes and, on the whole, being more likely to contribute to the research excellence framework (REF: this is the process through which university research is assessed in order to determine the future level of government funding). We would encourage funders to address this issue in order to protect their investment in these researchers.

Some of our respondents were extremely critical about these matters. "Career progression is very non-transparent" said one, "Vague descriptions of the areas in which excellence is required, but no idea of the level equivalent to excellence. Getting a proleptic appointment is very difficult." A second commented as follows: "It is widely believed that if you have funded your own salary from grants for seven years then the school should take you on as a full-time lecturer. However this does not appear to be written down anywhere and may have been inconsistently applied."

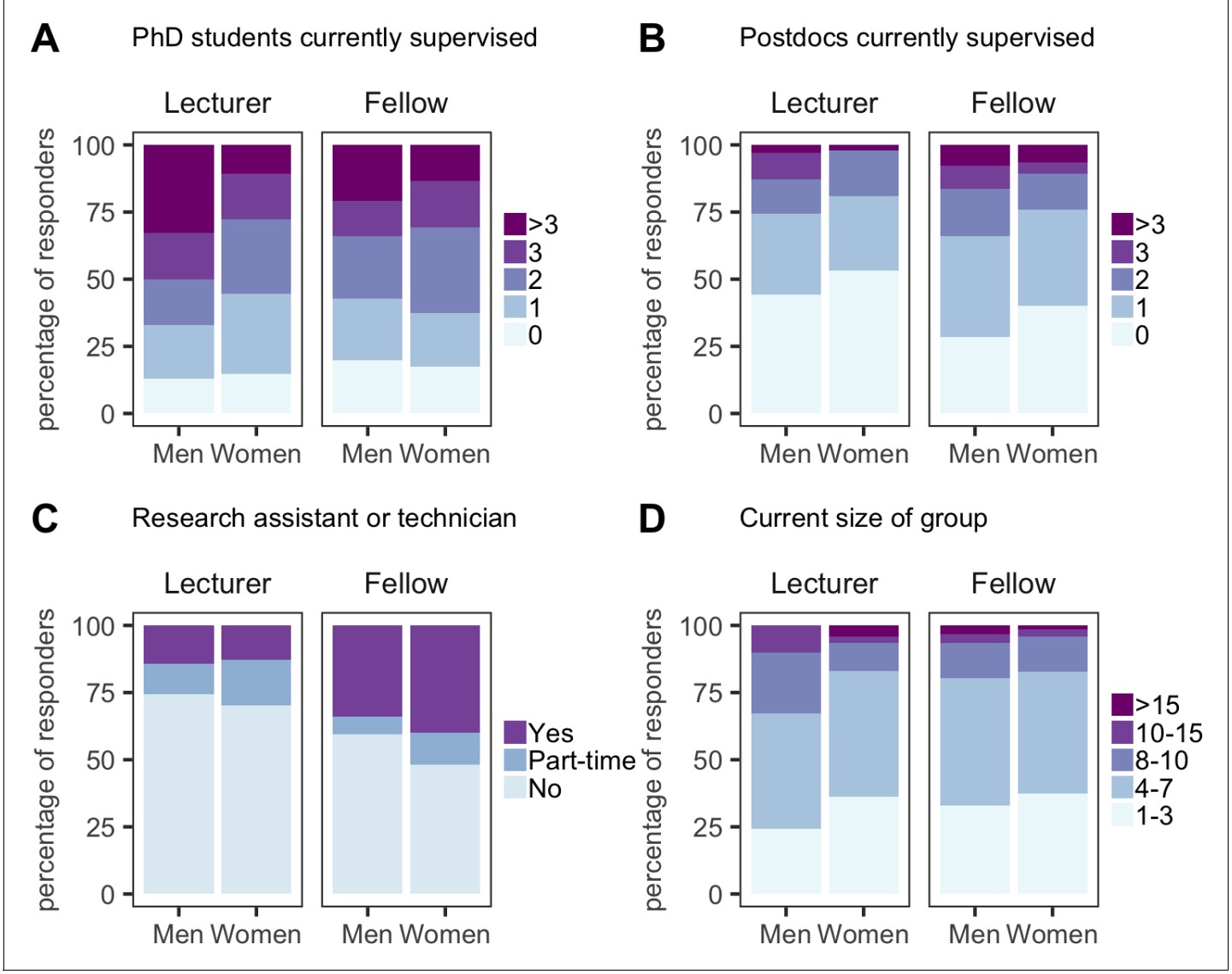

**Figure 9. Building a research group.** PhD students currently supervised (**A**), postdocs currently supervised (**B**), presence of a research assistant or technician (**C**) and current size of research group (**D**) for lecturers (men and women) and research fellows (men and women). Responders were asked to include undergraduates and master's students when reporting the size of their research group. All categories are expressed as the percentage of respondents within each category.

DOI: https://doi.org/10.7554/eLife.46827.028

The following source data is available for figure 9:

**Source data 1.** Summary data for *Figure 9*.

DOI: https://doi.org/10.7554/eLife.46827.029

### Building a research group

Although 80% of respondents had at least one PhD student (*Figure 9A*), both lecturers and research fellows reported difficulty in recruiting PhD students. One reason was that many PhD studentships in the UK have been gathered into large doctoral training centres, and senior researchers have been more successful in having their projects accepted by these centres: moreover, students seem to have a preference for joining established labs.

Some 80% of research fellows also had at least one postdoc in their lab, and some had three or more: however, 50% of lecturers did not have a postdoc (*Figure 9B*). Research

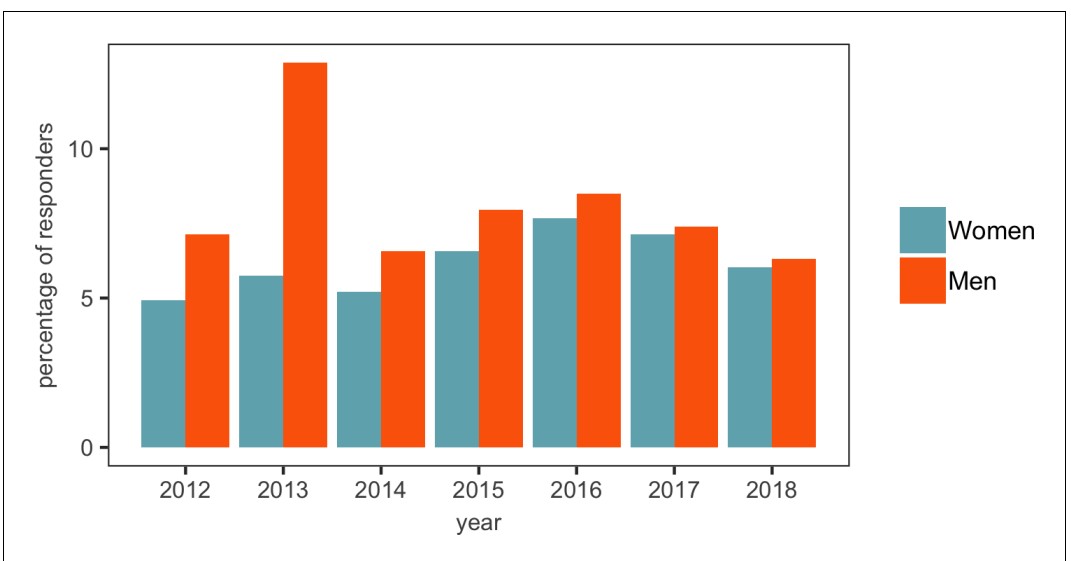

**Figure 10.** Recruitment of men and women by year. The gap between the number of men and women appointed as new PIs seems to have narrowed in recent years (with the gap being eight in 2012 and just one in 2017 and 2018), with the very noticeable exception of 2013, when 47 men and 21 women were appointed. A possible explanation for this is discussed in the text.

DOI: https://doi.org/10.7554/eLife.46827.030

The following source data is available for figure 10:

**Source data 1.** Summary data for **Figure 10**.

DOI: https://doi.org/10.7554/eLife.46827.031

fellows were also more likely to have a research assistant than lecturers (53% vs 25%; **Figure 9C**). However, lecturers tended to have bigger groups than research fellows (**Figure 9D**), which suggests high numbers of undergraduate students in the lab: in the absence of postdocs or research assistants to help train and supervise these students, this will only add to the pressure on the new PI, and may not provide the best training for the students either.

Again, the best way to convey the frustrations experienced by some respondents is to quote them directly: "To be successful as a fellow it is primordial to get a PhD student during the first year of contract," said one. "Without hands in the lab we cannot work. This is not granted, I struggled to get my lab members. Actually I secured an external studentship, but incredibly and annoyingly my Institution does not allow me to be primary supervisor". A second commented as follows: "I was told in no uncertain terms that the department could offer me nothing as a start-up. I am part of 2 possible PhD schemes in the university but funding only has been awarded to senior colleagues."

*Gender bias in recruitment*

As reported in **Figure 1B**, 56.7% of respondents were men, 41.6% were women and 1.6% preferred not to say. However, when we plot the percentage of male and female respondents against year of appointment, we find a spike in the number of male researchers appointed as new PIs in 2013 (**Figure 10**). We feel it is likely that this spike was caused by a wave of recruitment ahead of REF2014. The next REF will be in 2021. Each REF tends to be preceded by a 'transfer window' in which new academic staff are appointed and established staff sometimes move between universities. However, it is extremely clear from our data that the wave of new appointments before REF2014 significantly favoured male applicants. It seems likely to us that this wave of male recruitment may have been due to an increase in direct head-hunting, or more informal recruitment techniques driven by networks within fields, and that these practices seem to favour men. We urge institutions to ensure that female academics are not disadvantaged in this way ahead of REF2021.

## Recommendations to improve support for new PIs

While the career trajectories of academics are diverse, our survey suggest that there are a number of overarching issues that affect most new PIs. We list below actions that could be taken by host institutions and funders to address these overarching issues and ensure that all new PIs have the opportunity to reach their full potential.

### Actions by host institutions

- Recruitment should be driven by long-term strategy based around an investigator's potential. The recruiting department should conduct a rigorous interview for both fellowship candidates and lectureships with a long-term view to support that individual.
- Ensure that all new lecturers and research fellows are appointed a departmental mentor.
- Ensure that all new lecturers and research fellows receive a formal annual appraisal.
- Ensure that the criteria for promotion and proleptic appointments are transparent, and that these criteria are communicated to new lecturers and research fellows when they are appointed.
- Arrange that all research fellows be assessed for a proleptic appointment, or supported in their application for a senior fellowship, at least 24 months before the end of their fellowship (year 3/4).
- Have a policy of non-negotiable start-up packages for all recruits to help avoid gender bias.
- Ensure oversight of starting salaries by human resources to track and eliminate gender bias.
- Limit the number of undergraduate and masters project students supervised by a new PI to fewer than the number of lab members able to provide supervision (ie, PhD students, postdocs, RAs and the PI).
- Include new PIs in university-administered doctoral training programmes and/or award a proportion of PhD studentships to new PIs.
- Ensure that research fellows can spend the majority of their time on research and do not have a significant teaching load.
- Consider a standard policy that new lecturers and university-funded fellows be appointed at grade 8 and considered for promotion to senior lecturer (grade 9) upon successfully winning their first major research grant.
- Consider a standard policy that research fellows be appointed at grade 9 if they start their position with a major external research grant.

### Action by funders

- Reconsider the decision to fund PhD studentships primarily through large university-administered training programs as this approach can favour established labs.
- Consider including PhD studentships in fellowship awards.
- Funders should withhold funds from the host institutions if commitments such as lab space or access to facilities is not provided.
- Funders should engage with host institutions to monitor the career progression of research fellows, to ensure equal and fair assessment of fellows and lecturers.
- Consider a standard policy to recommend that research fellows be appointed at the equivalent of a senior lecturer.

### Advice when applying for new PI positions

- When visiting an institution that might employ you as a new PI, don't leave anything to chance when discussing what taking up a position at this institution would involve.
- Be aware that you are being recruited to become part of a department, so you should fully understand the department's goals and what role you are expected to fill as you join.
- Talk details. Ask to see the lab space you will be working in; ask who will provide lab basics like the fridges and freezers; find out what administrative support you will have (ordering, finance, travel bookings).
- Talk starting grade/salary, because where you start in the system will impact your future promotions. If you are being appointed as a new lecturer in the UK, negotiate to be appointed at grade 8, moving to grade 9 when you get your first big grant. If you are being appointed as a research fellow and are bringing a big grant with you, negotiate to be appointed as grade 9.
- Ask to speak with other new PIs, either in your department or in other departments, to find out how the host institution works and how they were recruited and supported.
- Make sure you have a mentor in your department.

- Make sure you have an annual review, and that you know the criteria you will need to meet to be promoted (or awarded a proleptic appointment).
- Don't assume that the person you are negotiating with has the power or authority to agree to what you are requesting, and be aware that heads of department can change, so be sure to get everything in writing.
- Once you start as a new PI, find your peers and talk with them often. Starting a lab can be a lonely business and is very different from being a postdoc, so other new PIs will be your best support network.

## Methods

The survey was conducted using convenience sampling, with most participants finding out about it through Twitter or forwarded email invitations. The majority of those responding were in the life sciences, in part because of the networks that the survey was circulated around, and partly due to the language used in the survey (the term PI does not have the same meaning in the social sciences, for instance). We do not claim that this is a fully representative sample. However, the sample has a similar gender and nationality breakdown to those reported for the sciences elsewhere (e.g., by the Higher Education Statistics Agency in the UK), and we do feel that it allows us to say important things about at least a significant subgroup of new PIs in the UK.

A pilot version of the survey was originally run with 10–12 volunteers recruited via the UK_NewPI slack group. On the basis of this a number of questions were revised, including changing some definitions, revising categories, and changing the salary question to refer to monetary values rather than pay bands. The revised survey with full questions can be found in *Supplementary file 2*. The survey was distributed, through the authors networks, via the NewPI slack (both UK and US), the eLife ambassadors mailing list, LinkedIn, and Twitter. In the end, of the 365 respondents, 311 were recruited via Twitter, 11 via email links, 16 via weblinks, and 1 from LinkedIn.

Much of the analysis consists of simple descriptive statistics – that is, looking at the distribution of individual variables. However, where we were interested in the relationship between variables, we used a mix of ordinal logit regressions and chi-squared tests depending on the nature of the relationship being studied (see *Supplementary file 1* for details). Ordinal

regression allowed us to control for multiple factors, to be more sure that the relationship that we found was not a result of (at least measured) confounding factors, for ordinal outcome variables. Full details of these models (both model estimates and specific variables used in different models) can be found in *Supplementary file 1*. The models used were as follows:

i. Ordinal logit regressions, with starting salary (8 bands) as the outcome variable, and gender as the main predictor variable of interest. Control variables include ethnicity (white/non-white), nationality (UK born/not), age, year of appointment, years between PhD and appointment, number of children, and type of appointment (lecturer/research fellow). These models allow us to see what individual factors are most associated with higher and lower salaries on appointment.

ii. Ordinal logit regressions, with 'optimism for the future' as the outcome variable (5-point likert scale). Various combinations of covariates were used including gender, ethnicity, nationality, whether respondent does teaching, starting salary (8 bands) number of grants received, value of grants received, whether respondent is in a Russell group university, whether the respondent has an RA/Tech (3 categories: Own RA, shared RA, no RA), number of PhD students, number of postdocs, years between PhD and PI appointment, year of PI appointment, and type of job (lecturer/fellow). These models allowed us to see what individual factors are most associated with feeling optimistic/pessimistic about their future careers.

iii. Ordinal logit regressions, with 'optimism for the future' as the outcome variable (5-point likert scale). Predictor variables were satisfaction with the lab, the department and the university/institute. The purpose of these models was to show which of those three influenced the respondents' optimism the most.

iv. A Chi squared test of REF year (2013 against all other years) and gender. This revealed the increase in male (but not female) recruitment associated with the REF.

v. Chi Squared test of type of job (lecturer, senior lecturer, research fellow and other) against amount of teaching (6 categories) vi) An ordinal logit model where the outcome is grant income (5 categories) and the outcome of interest is gender, with control variables of ethnicity

(white/non-white), age, year started as a PI, years of postdoctoral experience, number of children, and the year of first appointment as a PI. We also ran the same model as a standard logit model whether the outcome was dichotomised (over one million/less than one million).

## Acknowledgements
We thank all the members of UK_NewPI slack group for active discussion, honesty and productive practical suggestions for improvement which could be made to ease this difficult career stage. Email one of the authors if you would like to join this group. Thank you also to Dr Yanlan Mao (MRC LMCB, University College London) and Dr Ian Sudbery (University of Sheffield) for input on the structure and content of the survey questions, and to the MRC LMCB Athena Swan and EDIC committee for their discussions and engagement in this project. Thank you the #eLI-FEAmbassadors who have supported this project and thank you to the funders and universities who have engaged with us throughout the analysis and are enthusiastic to listen and to make change.

**Sophie E Acton** is a Cancer Research UK career development fellow in the MRC Laboratory for Molecular Cell Biology, University College London, United Kingdom
s.acton@ucl.ac.uk
https://orcid.org/0000-0003-2704-716X

**Andrew JD Bell** is a lecturer in the Sheffield Methods Institute, University of Sheffield, United Kingdom
andrew.j.d.bell@sheffield.ac.uk
https://orcid.org/0000-0002-8268-5853

**Christopher P Toseland** is an MRC Career Development awardee in the Department of Oncology and Metabolism, University of Sheffield, United Kingdom
c.toseland@sheffield.ac.uk
https://orcid.org/0000-0002-1641-7535

**Alison Twelvetrees** is a Vice Chancellor's fellow in the Sheffield Institute for Translational Neuroscience, University of Sheffield, United Kingdom
a.twelvetrees@sheffield.ac.uk
https://orcid.org/0000-0002-1796-1508

*Author contributions:* Sophie E Acton, Christopher P Toseland, Conceptualization, Data curation, Investigation, Writing—original draft, Writing—review and editing; Andrew JD Bell, Software, Formal analysis, Validation, Methodology, Writing—review and editing; Alison Twelvetrees, Conceptualization, Data curation, Visualization, Writing—original draft, Writing—review and editing

*Competing interests:* The authors declare that no competing interests exist.

*Ethics:* Human subjects: Participants were invited to respond to the survey under the understanding that only summary data would be used or presented. We collected no identifying information and as such all responses are anonymous. Having sought the advice of the UCL ethics committee we are informed that our study is exempt from the need for ethics approval under the following category: "Research involving the use of non-sensitive, completely anonymous educational tests, survey and interview procedures when the participants are not defined as 'vulnerable' and participation will not induce undue psychological stress or anxiety."

**Decision letter and Author response**
Decision letter https://doi.org/10.7554/eLife.46827.036
Author response https://doi.org/10.7554/eLife.46827.037

## Additional files
### Supplementary files
• Supplementary file 1. Statistical analysis.
DOI: https://doi.org/10.7554/eLife.46827.032
• Supplementary file 2. Survey questionnaire.
DOI: https://doi.org/10.7554/eLife.46827.033

## Data availability
Summary data analysed in this study are included in the manuscript and supporting files.

