## [Decision Letter]

Thank you for submitting your article "The life of P.I. - Transitions to Independence in Academia" for consideration by eLife. Your article has been reviewed by 3 peer reviewers, and the evaluation has been overseen by myself (Emma Pewsey) and the eLife Features Editor, Peter Rodgers. The following individuals involved in review of your submission have agreed to reveal their identity: Amanda N Sferruzzi-Perri (Reviewer #1).

The reviewers have discussed the reviews with one another and the Reviewing Editor has drafted this decision to help you prepare a revised submission. While the list of points to address may seem long and will require major revision of the article, we feel that they will greatly benefit the clarity of the article, and its ability to influence decision-makers. We hope you will be able to submit the revised version within two months.

SUMMARY OF MANUSCRIPT

This is an important and timely topic that should get major spotlight and we congratulate the authors for deciding to tackle it. The paper does a good job of identifying the major struggles of new PIs, and makes some suggestions for how PIs, host institutions and funding agencies can help to prevent these struggles. The opportunity that the findings present to influence senior academics and other decision makers therefore make it crucial that this manuscript is of excellent quality and that it leads to clear conclusions about why and how the situation should change.

LIST OF REQUIRED REVISIONS

GENERAL

1. There are a number of incorrect citations (for example, the in-text citations for the panels of Figure 10 are labelled as Figure 9, and a citation to 14F appears to be for 14G). Also, the 4th sentence in the caption to figure 13 should start: "Fellows (D) and Lecturers (E) were then.. . ". Please check and correct the text and figure captions for issues like these. Please also ensure that all in-text discussion of data that's presented in a figure cites that figure.

2. Please present the findings in a more neutral manner. For example, instead of making a broad statement such as "too many new investigators express frustration", please report on the percentages who are frustrated.

PRESENTATION OF FIGURES

3. The article will be easier to read if the number of figures is reduced. eLife allows the figures in a paper to have figure supplements; the figures appear in both the online and pdf versions; the figure supplements only appear in the online version, but the figure caption makes it clear that the figure supplement are available online. Please combine figures and convert some panels into figure supplements as suggested below.

4. Some panels are not discussed in the text (2C, 6A, 6B, 6D, 6E, 7A, 7B, 9C, 10A, 13B, 13C, 14D, 14E, 14F, 14I, 15A); please either i) delete these or ii) convert them to figure supplements (and make sure they are cited somewhere in the text or the relevant figure caption).

5. The results of yes/no questions (5A, 5B, 5C, 6C, 10B, 10C, 10D, 10E, 12A, 12B, 13A, 14A, 14B) would be better presented by stating the percentage of respondents who answered yes in the text and/or figure caption, and citing the appropriate section of your supporting data file.

6. Please include y-axes and keys for figures where these are missing (including Figures 3, 4, 11).

7. Figure panels 14G and H show the same data, so please convert one to a figure supplement. Also, for 14G, please adjust the y-axis scaling to make it possible to see the data that supports the statement in the text about start-up funds "in the region of £20-60K".

8. Please ensure that the order of the figures matches the order in which they are cited/mentioned in the text.

RESULTS

9. Please present the results of the statistical tests for all the data shown; for example, this is missing from comparisons between female and male PIs and/or lecturers and researcher fellows. Please state the types of statistical tests run for each piece of presented data, with details on how certain confounders were controlled for, as well as detailing any corresponding statistical outcomes.

10. The first sentence of the results states that '...365 respondents represents a significant proportion of new group leaders in the UK.' To support this comment, please state the number of new group leaders in the UK for the period corresponding to the results presented.

11. Please rephrase the final sentence for the second paragraph of the results ('There has been some discussion... in redressing the balance') to make its meaning clearer.

12. Please clarify what Figure 2B represents by describing it in the figure legend; you may also wish to add further clarification in the main text. Should the values for women and men add to 100%? Does this graph show only those that had dependents/caring responsibilities? Does '0' for length of break mean the individual had a dependent but took 0 months off from work? What is the mean and median for the data for each gender? How does this relate to the number of dependents per respondent? Furthermore, the panel shows possibly 1 or 2 men taking very long periods of leave, which does not appear to support the statement in the text that '... fathers are also taking significant periods of leave and sharing childcare responsibilities.' Please clarify.

13. Please provide the data supporting the following sentence: 'It was also encouraging to find that having dependents did not affect satisfaction ratings or optimism scores for either females or males'; the data presented is not separated by sex.

14. Instead of stating that the proportion of lecturers to research fellows captured in the study was roughly equal, please state the exact percentages.

15. The comments about research fellows being based at Russell group universities (bottom of page 3) seems to reflect that these universities first recruit individuals as research fellows (rather than as lecturers), as shown in Figure 5D. Please discuss this in the text.

16. Could the data presented in Fig 8 and 15 also relate to when career breaks may occur for a women (which we know are for longer periods)? Was information collected about the timing of leave with regard to the trajectory of the respondent's career paths? In regards to 8C, could this be related to men asking for more money for each grant compared to women? Could a calculation be made to determine whether the amount awarded per grant varied between women and men?

17. Based on the data presented in 9B and C, women have the larger loads for teaching and committees. Please discuss this, as these data may be related to the finding that women also secure fewer grants and have small lab sizes - as they are simply over-committing themselves to, or are more commonly tasked with, non-research activities.

18. The sentence 'New PIs should also be aware that supervision of master's students and PhD students would also count as teaching contribution', is not strictly true for all universities. There are differences in what universities classify as part of a teaching load.

19. Please check that the data presented in 10C does correspond to the sentence stating 'Nearly 18% of externally funded fellows reported not having an.. .'.

20. The statement '...participated by submitting projects to up to 5-6 in some cases (Fig 14C).' does not agree with the data shown in Figure 14C.

21. Please show the data that supports the statement that '...lecturers were the least satisfied with worklife balance compared to research fellows'.

RECOMMENDATIONS FOR IMPROVED SUPPORT OF NEW PRINCIPAL INVESTIGATORS

22. Many of the advice points are not related to issues arising from the data. It would instead be more compelling to discuss recommendations directly related to the data and/or to discuss sources that may support the potential effectiveness of these solutions. The article would also be strengthened by more recommendations that new PIs can act on regardless of host support. For example when mentioning that mentorship to new PIs is not ideal, you could direct readers to resources for effective use of mentorship.

23. For the actions by host institutions, could you recommend anything related to the observations that there are clear gender differences in salary and start-ups, as well as recruitment around the REF?

24. Please explain the rationale for suggesting different grades for research fellows and university-funded fellows (action points 7 and 8 for host institutions).

METHODS AND SUPPORTING DATA

25. Please expand the methods section to give full details of the nature of the items included in the survey and explain why these were chosen, the pilot testing and revision process, the specific definition of the sought respondents, and the recruitment strategy used. Please also refer the reader to the document of the final survey questions.

26. Please provide a supporting data file that clearly states the number and percentages of respondents to each option in the survey, broken down by gender and lecturer/fellow as appropriate.

[Editors' note: further editorial revisions were requested prior to acceptance, and these revisions resulted in changes to the numbering of the figures.]

---

## [Author Response]

We repeat the reviewers’ points here in italic, and include our replies point by point, as well as a description of the changes made, in Roman.

GENERAL

1. There are a number of incorrect citations (for example, the in-text citations for the panels of Figure 10 are labelled as Figure 9, and a citation to 14F appears to be for 14G). Also, the 4th sentence in the caption to figure 13 should start: "Fellows (D) and Lecturers (E) were then.. . ". Please check and correct the text and figure captions for issues like these. Please also ensure that all in-text discussion of data that's presented in a figure cites that figure.

REPLY: The figure panels and numbering have all been updated to address edit suggested below. The text has been edited to reflect all of these changes. Many panels have also been removed from the main figures to streamline the manuscript, and now form supplements for each main figure.

2. Please present the findings in a more neutral manner. For example, instead of making a broad statement such as "too many new investigators express frustration", please report on the percentages who are frustrated.

REPLY: We have edited the text with this comment in mind, all changes are tracked in the submitted text document. We have deleted statements such above, and replaced with numbers from the data collected.

PRESENTATION OF FIGURES

3. The article will be easier to read if the number of figures is reduced. eLife allows the figures in a paper to have figure supplements; the figures appear in both the online and pdf versions; the figure supplements only appear in the online version, but the figure caption makes it clear that the figure supplement are available online. Please combine figures and convert some panels into figure supplements as suggested below.

REPLY: We have rearranged all of the figures to streamline the manuscript

Please see table of rearranged figures below

4. Some panels are not discussed in the text (2C, 6A, 6B, 6D, 6E, 7A, 7B, 9C, 10A, 13B, 13C, 14D, 14E, 14F, 14I, 15A); please either i) delete these or ii) convert them to figure supplements (and make sure they are cited somewhere in the text or the relevant figure caption).

REPLY: Please see table of rearranged figures below

5. The results of yes/no questions (5A, 5B, 5C, 6C, 10B, 10C, 10D, 10E, 12A, 12B, 13A, 14A, 14B) would be better presented by stating the percentage of respondents who answered yes in the text and/or figure caption, and citing the appropriate section of your supporting data file.

REPLY: This is now changed. Please see table of rearranged figures below for updated figure numbering.

6. Please include y-axes and keys for figures where these are missing (including Figures 3, 4, 11).

REPLY: This is now changed. Please see table of rearranged figures below for updated figure numbering.

7. Figure panels 14G and H show the same data, so please convert one to a figure supplement. Also, for 14G, please adjust the y-axis scaling to make it possible to see the data that supports the statement in the text about start-up funds "in the region of £20-60K".

REPLY: Please see table of rearranged figures below

8. Please ensure that the order of the figures matches the order in which they are cited/mentioned in the text.

REPLY: Please see table of rearranged figures below

Figure in manuscriptReviewer suggested actionsFinal Figure numberingFigure 1A - genderFigure 1BFigure 1B - nationalityFigure 1CFigure 1C - ageDELETEDFigure 1D - raceFigure 1DFigure 1E - dependentsFigure 1-figure supplement 2AFigure 1F - fieldFigure 1AFigure 2A - postdoc experienceFigure 1EFigure 2B - career breaksFigure 1-figure supplement 2BFigure 2C - age at independencediscuss or deleteFigure 1-figure supplement 1Figure 3 - mobilityFigure 2Figure 4 - satisfactionFigure 3Figure 5A - full time/part timedeleteFigure 1 - figure supplement 2CFigure 5B - uni or instdeleteDELETEDFigure 5C - russelldeleteDELETEDFigure 5D - first jobFigure 4AFigure 6A - advertised portdiscuss or deleteDELETEDFigure 6B - position createddiscuss or deleteDELETEDFigure 6C - moved to start groupdeleteFigure 2 - figure suplement 1Figure 6D - previous in current deptdiscuss or deleteDELETEDFigure 6E - probation perioddiscuss or deleteDELETEDFigure 7A - £ requireddiscuss or deleteFigure 4BFigure 7B - external fellowhsipdiscuss or deleteDELETEDFigure 7C - external fundingFigure 4CFigure 8A - grant success by yearFigure 5AFigure 8B - grants recievedFigure 5BFigure 8C - grant incomeFigure 5CFigure 8D - starting salaryFigure 5DFigure 9A - teachingFigure 6AFigure 9B - contact hoursFigure 6BFigure 9C - committeesdiscuss or deleteFigure 6CFigure 10A - dept sizediscuss or deleteDELETEDFigure 10B - mentordeleteDELETEDFigure 10C - annual reviewdeleteDELETEDFigure 10D - union memberdeleteDELETEDFigure 10E - fellows meetingsdeleteDELETEDFigure 11 - mentoringFigure 7Figure 12A - proleptic appointmentdeleteDELETEDFigure 12B - proleptic in placedeleteDELETEDFigure 13A - clear criteriadeleteDELETEDFigure 13B - promotion criteriadiscuss or deleteDELETEDFigure 13C - proleptic offereddiscuss or deleteFigure 8 - figure supplement 1AFigure 13D - fellows porlepticFigure 8 - figure supplement 1BFigure 13E - lecturerers promotionDELETEDFigure 14A - PhD studentdeleteDELETEDFigure 14B - PhD programmesdeleteDELETEDFigure 14C - number of programmesFigure 8 - figure supplement 2Figure 14D - facility accessdiscuss or deleteDELETEDFigure 14E - lab spacediscuss or deleteDELETEDFigure 14F - start up fundsdiscuss or deleteDELETEDFigure 14G - start up amountpoint 7DELETEDFigure 14H - start up amountpoint 7Figure 8Figure 14I - lab space fundsdiscuss or deleteDELETEDFigure 15A - PhDs to completiondiscuss or deleteDELETEDFigure 15B - PhD studentsFigure 9AFigure 15C - postdocsFigure 9BFigure 15D - RA/techFigure 9CFigure 15E - group sizeFigure 9DFigure 16Figure 10NEWrespond to point 16NEW Figure 1 - figure supplement 2DNEWresponding to point 21NEW Figure 9 - figure supplement 1NEWto go with "It was also encouraging to find that having dependants did not affect satisfaction ratings or optimism scores for either male or female PIs."NEW Figure 3 - figure supplement 1NEWGrade level appointed at - was previously missingNEW Figure 5 - figure supplement 1BNEWStarting salary for Lecturere's Fellow - see CT comment on line 201NEW Figure 5 - figure supplement 1A

RESULTS

9. Please present the results of the statistical tests for all the data shown; for example, this is missing from comparisons between female and male PIs and/or lecturers and researcher fellows. Please state the types of statistical tests run for each piece of presented data, with details on how certain confounders were controlled for, as well as detailing any corresponding statistical outcomes.

REPLY: The methods section has been extended and updated, changes are tracked in the attached document. We now also include. xls files for each graph shown, which makes clear the total numbers for each category/response.

10. The first sentence of the results states that '...365 respondents represents a significant proportion of new group leaders in the UK.' To support this comment, please state the number of new group leaders in the UK for the period corresponding to the results presented.

REPLY: We have been unable to find a total number of PIs in the UK, this statement was based on estimates knowing the approximate number of fellowships awarded by each of the larger funding bodies per cycle. However, we would be happy to remove this statement if required.

11. Please rephrase the final sentence for the second paragraph of the results ('There has been some discussion... in redressing the balance') to make its meaning clearer.

REPLY: Please see tracked changes in the attached text.

12. Please clarify what Figure 2B represents by describing it in the figure legend; you may also wish to add further clarification in the main text. Should the values for women and men add to 100%? Does this graph show only those that had dependents/caring responsibilities? Does '0' for length of break mean the individual had a dependent but took 0 months off from work? What is the mean and median for the data for each gender? How does this relate to the number of dependents per respondent? Furthermore, the panel shows possibly 1 or 2 men taking very long periods of leave, which does not appear to support the statement in the text that '... fathers are also taking significant periods of leave and sharing childcare responsibilities.' Please clarify.

REPLY: Figure 2 B has been edited and now forms part of the supplement for figure 1. We do not have data for the number of dependents per respondent, this was sometimes supplied in the comments but we are unable to break the data down in this way.

13. Please provide the data supporting the following sentence: 'It was also encouraging to find that having dependents did not affect satisfaction ratings or optimism scores for either females or males'; the data presented is not separated by sex.

REPLY: This data is now included as a supplement to figure 3, split by gender as requested.

14. Instead of stating that the proportion of lecturers to research fellows captured in the study was roughly equal, please state the exact percentages.

REPLY: This has now been addressed

15. The comments about research fellows being based at Russell group universities (bottom of page 3) seems to reflect that these universities first recruit individuals as research fellows (rather than as lecturers), as shown in Figure 5D. Please discuss this in the text.

REPLY: This has now been addressed, please see tracked changes

16. Could the data presented in Fig 8 and 15 also relate to when career breaks may occur for a women (which we know are for longer periods)? Was information collected about the timing of leave with regard to the trajectory of the respondent's career paths? In regards to 8C, could this be related to men asking for more money for each grant compared to women? Could a calculation be made to determine whether the amount awarded per grant varied between women and men?

REPLY: We did not collect data regarding the timing of leave with the career trajectory, although this would be an interesting question. We also did not ask for grant amount per application, and since this would vary greatly depending on the type of grant it would not be possible to draw concrete conclusions. It would be very interesting to ask funders to look at the grant amount awarded split male to female.

17. Based on the data presented in 9B and C, women have the larger loads for teaching and committees. Please discuss this, as these data may be related to the finding that women also secure fewer grants and have small lab sizes - as they are simply over-committing themselves to, or are more commonly tasked with, non-research activities.

REPLY: This has now been addressed, please see tracked changes

18. The sentence 'New PIs should also be aware that supervision of master's students and PhD students would also count as teaching contribution', is not strictly true for all universities. There are differences in what universities classify as part of a teaching load.

REPLY: We have edited this please see tracked changes

19. Please check that the data presented in 10C does correspond to the sentence stating 'Nearly 18% of externally funded fellows reported not having an.. .'.

REPLY: This figure has now been removed from the manuscript

20. The statement '...participated by submitting projects to up to 5-6 in some cases (Fig 14C).' does not agree with the data shown in Figure 14C.

REPLY: This data has been move to Figure 8 - figure supplement 2, and the text has been edited to match the plots.

21. Please show the data that supports the statement that '...lecturers were the least satisfied with worklife balance compared to research fellows'.

REPLY: This is now shown in a new figure supplement - NEW Figure 9 - figure supplement 1

RECOMMENDATIONS FOR IMPROVED SUPPORT OF NEW PRINCIPAL INVESTIGATORS

22. Many of the advice points are not related to issues arising from the data. It would instead be more compelling to discuss recommendations directly related to the data and/or to discuss sources that may support the potential effectiveness of these solutions. The article would also be strengthened by more recommendations that new PIs can act on regardless of host support. For example when mentioning that mentorship to new PIs is not ideal, you could direct readers to resources for effective use of mentorship.

REPLY: Please see tracked changes, we note that peer-peer mentorship can be extremely important and an excellent source of support.

23. For the actions by host institutions, could you recommend anything related to the observations that there are clear gender differences in salary and start-ups, as well as recruitment around the REF?

REPLY: Having discussed this with our host departments this is a very difficult issue to address, as they require a degree of flexibility in starting salary in order to recruit their favoured candidates who may have other offers.

24. Please explain the rationale for suggesting different grades for research fellows and university-funded fellows (action points 7 and 8 for host institutions).

REPLY: University funded fellowships are generally a scheme to bring in potential candidates for career development fellowships, so within our peer group this is generally viewed as a stepping stone towards independence which is solidified when external funding is awarded.

METHODS AND SUPPORTING DATA

25. Please expand the methods section to give full details of the nature of the items included in the survey and explain why these were chosen, the pilot testing and revision process, the specific definition of the sought respondents, and the recruitment strategy used. Please also refer the reader to the document of the final survey questions.

REPLY: Please see tracked changes, we have added detail to this section

26. Please provide a supporting data file that clearly states the number and percentages of respondents to each option in the survey, broken down by gender and lecturer/fellow as appropriate.

REPLY: We have uploaded the supporting data for each plot now shown, broken down as presented.